# Colostrum Features of Active and Recovered COVID-19 Patients Revealed Using Next-Generation Proteomics Technique, SWATH-MS

**DOI:** 10.3390/children10081423

**Published:** 2023-08-21

**Authors:** Iván Hernández-Caravaca, Carla Moros-Nicolás, Leopoldo González-Brusi, Mª José Romero de Ávila, Catalina De Paco Matallana, Pablo Pelegrín, María Ángeles Castaño-Molina, Lucía Díaz-Meca, Javier Sánchez-Romero, Laura Martínez-Alarcón, Manuel Avilés, Mª José Izquierdo-Rico

**Affiliations:** 1Department of Community Nursing, Preventive Medicine and Public Health and History of Science, Campus de Sant Vicent del Raspeig, University of Alicante, 03690 Alicante, Spain; ivan.hernandez@ua.es; 2Instituto Murciano de Investigación Biosanitaria Pascual Parrilla (IMIB), Campus de Ciencias de la Salud, 30120 Murcia, Spain; carla.moros@um.es (C.M.-N.); leopoldo.gonzalez@um.es (L.G.-B.); katy.depaco@gmail.com (C.D.P.M.); pablo.pelegrin@imib.es (P.P.); ldm460@hotmail.com (L.D.-M.); lma5@um.es (L.M.-A.); maviles@um.es (M.A.); 3Departamento de Biología Celular e Histología, Facultad de Medicina, Universidad de Murcia, Campus Mare Nostrum (CMN), 30120 Murcia, Spain; mjose.romero@um.es; 4Servicio de Obstetricia y Ginecología, Hospital Clínico Universitario Virgen de la Arrixaca, 30120 Murcia, Spain; angeles.castano@um.es; 5Departamento de Bioquímica y Biología Molecular “B” e Inmunología, Facultad de Medicina, Universidad de Murcia, Campus Mare Nostrum (CMN), 30120 Murcia, Spain; 6Departamento de Enfermería, Facultad de Enfermería, Universidad de Murcia, Campus Mare Nostrum (CMN), 30120 Murcia, Spain; 7Unit, Department of Surgery, Virgen de la Arrixaca University Hospital, 30120 Murcia, Spain

**Keywords:** breastfeeding, colostrum, SARS-CoV-2, COVID-19, proteomics

## Abstract

Colostrum performs nutritional, anti-inflammatory and anti-infective functions and promotes immune system formation and organ development. The new coronavirus, SARS-CoV-2, has generated concerns about viral transmission through human milk, with a lack of evidence about human milk’s protective effects against the infection. This study aimed at analyzing presence of the virus and at identifying the protein expression profile of human colostrum in active and COVID-19-recovered patients. Colostrum samples were collected from women with COVID-19 (*n* = 3), women recently recovered from the infection (*n* = 4), and non-infected women (*n* = 5). The samples were analyzed by means of RT-qPCR to determine presence of the virus and using SWATH-MS for proteomic analysis. Proteomic results were then analyzed using bioinformatic methods. The viral tests were negative for SARS-CoV-2 in the colostrum from COVID-19 patients. The proteomic analysis identified 301 common proteins in all samples analyzed. Nineteen proteins were upregulated and 7 were downregulated in the COVID-19 group versus the control samples, whereas 18 were upregulated and 7 were downregulated when comparing the COVID-19 group to the recovered group. Eleven proteins were biomarkers of active COVID-19 infection. Ten were upregulated: ACTN1, CD36, FAM3B, GPRC5B, IGHA2, IGK, PLTP, RAC1, SDCBP and SERPINF1, and one was downregulated: PSAP. These proteins are mainly related to immunity, inflammatory response and protein transport. In conclusion, the results of this study suggest that colostrum is not a vehicle for mother-to-child SARS-CoV-2 transmission. Moreover, the colostrum’s proteome of active and recuperated patients indicate that it could provide immune benefits to infants.

## 1. Introduction

Coronavirus (CoV) is a family of single-stranded RNA viruses that are classified into four genera (alpha, beta, gamma and delta). Alpha and beta cause respiratory diseases in humans, namely: severe acute respiratory syndrome virus (SARS-CoV-1) and Middle East respiratory syndrome virus (MERS-CoV). SARS-CoV-2 is a novel beta CoV that emerged in late 2019 causing coronavirus disease 2019 (COVID-19) [1]. Thus, the World Health Organization (WHO) declared this disease (COVID-19) a pandemic on 11 March 2020. SARS-CoV-2 has caused more than 500 million confirmed COVID-19 cases, resulting in more than 6 million deaths worldwide up to 27 April 2022 (WHO Coronavirus (COVID-19) Dashboard|WHO Coronavirus (COVID-19) Dashboard with Vaccination Data, n.d.) [2]. Based on data from the United States, more than 200,000 pregnant women have been infected with SARS-CoV-2 [3]. Concerns about the SARS-CoV-2 transmission risk and the lack of evidence for the protective effects of human milk against the virus have produced a debate on whether mothers with COVID-19 should breastfeed or not. Transmission of the virus is by aerosols and close contact, which is why global health measures were adopted to prevent transmission, including social distancing. This has led to the separation of mothers and newborns in cases of COVID-19-positive or suspected mothers, avoiding close contact between mother and newborn, thus hindering breastfeeding [4].

Human milk contains nutrients and bioactive molecules to promote growth and development of the newborn [5]. It influences the short- and long-term health of the infant. In the short term, breastfeeding is associated with a reduction in infections among infants, which reduces the hospitalization risk [6]. According to the WHO, the main long-term benefits of breastfeeding are as follows: increased performance on intelligence and cognitive development tests, significantly reduced risk of obesity in childhood and adulthood, reduced risk of type II diabetes and a possible cardioprotective effect [7]. The benefits for mothers of breastfeeding have been demonstrated by several studies showing that there is an inverse association between breastfeeding duration and breast and ovarian cancer [8].

Human milk gradually changes its composition over the days. Thus, different types of milk can be distinguished: colostrum, transitional milk and mature milk. Colostrum is considered the human milk produced from delivery to 72 h postpartum, transitional milk from 3 to 15 days, and mature milk more than 16 days [9]. When compared to mature milk, colostrum contains higher amounts of proteins, growth factors and fat-soluble vitamins. Although in lower concentrations, it also contains fat, lactose and water-soluble vitamins. It has high protein content and immunological factors and is rich in polyunsaturated fatty acids (n-3 and n-6) [10]. Colostrum contains high concentrations of immunoglobulins, especially secretory IgA, indicating that its main function is immunological and not nutritional. It transfers passive immunity to the newborn by intestinal absorption of immunoglobulins [11]. Colostrum intake by the newborn is of paramount importance because intestinal permeability to proteins with high molecular weight decreases around the first few days of life [12].

Breastfeeding is related to low rates of respiratory and gastrointestinal infections in childhood [13]. It is known that viral infections can produce changes in human milk composition, which might lead to modifications in neonatal immunity and metabolism [14]. For this reason, there are a few virus infections for which it is recommended that breastfeeding be discontinued or not initiated. They include infections with type I or II human T-cell lymphotropic virus, human immunodeficiency virus (HIV) or Ebola virus [15].

At birth, newborns are still immunologically immature. IgG is the only antibody that significantly passes through the human placenta [16]. Because of this, newborns’ protection depends on the mothers’ acquired passive immunity and on their own innate immunity. The immunoglobulins in human milk (sIgA, IgA, IgD, IgE, IgG and IgM,) exert the antimicrobial activity of human milk. The concentration of secretory IgA (sIgA) is 10–100 times higher in milk than in serum. sIgA modulates intestinal immunity and protects against several microorganisms. Therefore, these antibodies are produced by the mother and are a reflection of the bacteria and viruses that have entered into contact with the woman during the perinatal period. As in other infections, it has been reported that human milk from mothers infected with SARS-CoV-2 may also possess antibodies against SARS-CoV-2 [17].

Human milk also contains bioactive factors that inhibit inflammation and enhance the production of specific antibodies; compounds PAF acetylhydrolase; interleukins 1, 6, 8 and 10; secretory leukocyte protease inhibitors (SLPIs); antioxidants; transforming growth factor (TGF); and defensin-1 [18]. Moreover, it is known that human milk contains several antiviral proteins including lactoferrin, mucin-1 and secretory leucocyte protease inhibitor [17], although there is little information about human milk proteins in cases of SARS-CoV-2 infection.

The COVID-19 pandemic, and the responses to it, endangered essential elements for establishing and maintaining an effective maternal milk supply. Reduced breastfeeding due to the pandemic may lead to an increase in common infant disorders. This is due to fear of infection, initial confusion and clinical and economic constraints in health systems. Moreover, concerns about transmission of the virus and the lack of studies about human milk’s protection against SARS-CoV-2 have made it difficult for mothers to choose to breastfeed [19].

Given the above, the objectives of this study are as follows: 1. to analyze the presence of SARS-CoV-2 by means of RT-qPCR in the colostrum collected from mothers with COVID-19 and 2. to characterize the protein profile in the colostrum collected from mothers with COVID-19, post-infection recovered and non-infected donors using a new methodology: sequential window acquisition of all theoretical fragment ion spectra-MS (SWATH-MS). The study of the colostrum proteome in the different groups might help know if there are differences and whether these changes are maintained in time. This new methodology can provide direct information about the dynamics that human milk undergoes during and after COVID-19 infection. SWATH-MS has shown deep proteome coverage coupled to higher performance and reproducibility in previous studies when compared to conventional data-dependent analysis mass spectrometry (DDA-MS) [20,21].

## 2. Materials and Methods

### 2.1. Ethics

This study was approved by the CEIm Virgen de la Arrixaca (2021-3-4-HCUVA). All the donors involved in this study signed a written informed consent form and were not compensated for their participation in this study. Individual confidentially was maintained, a random participant code was assigned, and data were stored on a secure institutional server.

### 2.2. Research Design

This was a descriptive, observational, cross-sectional and quantitative study. Colostrum samples were collected 24 h after delivery from three groups of postpartum women, namely: (a) a group of 3 women with COVID-19 (positive PCR by nasal swab on delivery day); (b) a group of 4 COVID-19-recovered patients (they overcame the infection between 3 and 6 months before delivery and had negative PCR by nasal swab); and (c) a group of 5 noninfected women, or control group (PCR negative by nasal swab). The samples from the mothers with COVID-19 were analyzed by means of RT-qPCR to determine presence or absence of the virus in each sample. After protein extraction and digestion, the colostrum samples from all three groups were evaluated using theoretical fragment ion spectra-MS (SWATH-MS). Finally, statistical and bioinformatics analyses were performed to analyze the proteomic results (Figure 1).

### 2.3. Collection and Processing of Samples

Women with full-term pregnancies who had recently given birth were recruited for the study. Women were classified into three groups: with active COVID-19 infection, diagnosed before delivery and not active (PCR negative) and noninfected women. The inclusion and exclusion criteria are listed in Table 1.

The main characteristics of the colostrum donors were summarized in Table 2: maternal age, ethnicity, other infectious diseases, other complications during pregnancy, vaccination status against SARS-CoV-2, gravidity, type of delivery, gestational week at birth, postdelivery colostrum collection, and COVID-19 infection severity. The colostrum samples (929.2 ± 360.8 µL) were collected within 20 ± 5.9 h after vaginal delivery. Collection was by manual expression after hand washing in sterile containers. All samples were kept at −80 °C until analysis.

### 2.4. RT-qPCR

Total RNA was isolated using an RNeasy Mini Kit (Quiagen, Hilden, Germany) according to the manufacturer’s instructions. Secondly, each reaction was prepared with NZYTech Speedy One-Step RT-qPCR Probe Master mix and TaqMan CDC 2019-nCov RUO Kit probes (probes for N1 and RNAseP), E Assay_First line Screening (Idtdna, München-Flughafen, Germany).

The primers used for RT-qPCR for each sample are designed based on N1 gene and E gene sequences. Several controls were used, namely: 2019-CoV Plasmid Controls, containing the complete N1 nucleocapsid gene, and other positive plasmid controls for the E gene (envelope gene) (Table 3). Water was used as no template control.

RNAseP, which must be amplified with Ct ≤ 35, is used as quality control of the samples. RNA from HEK cells (RNA from any uninfected human cell culture) was also used as negative control.

### 2.5. Proteomic Analysis

Proteomic analyses were performed at the Central Support Service for Experimental Research at the University of Valencia Proteomics Unit, which is part of the Carlos III Health Institute ProteoRed Proteomics Platform.

#### 2.5.1. Sample Processing

The samples were processed as follows: 200 µL of every sample was centrifuged for 20 min at 15,000× *g* at 4 °C. Subsequently, a supernatant and a fat precipitate were separated from the intermediate aqueous fraction. One microliter of every aqueous fraction was quantified using Qubit (Invitrogen, Waltham, MA, USA) according to the manufacturer’s instructions, and sample reproducibility was checked by means of SDS-PAGE.

#### 2.5.2. Protein Digestion and Sample Preparation

A total of 40 µg of each individual sample was adjusted to 20 µL of 50 mM NH_4_HCO_3_ and reduced with 2 mM dithiothreitol (DTT) for 20 min at 60 °C. After cooling at room temperature (RT), cysteine residues were alkylated with 5.5 mM iodoacetamide (IAM) at RT for 30 min in the dark. Excess IAM was quenched using 10 mM DTT at 37 °C for 1 h. Trypsin (sequencing grade, Promega, Madison, WI, USA) was added at a 1:20 ratio and left to rest overnight at 37 °C. After digestion, the solutions were acidified with 0.1% formic acid. The final concentration of the peptide mixtures was 0.52 µg/µL. The proteins within the samples were separated in two groups: major proteins and minor proteins. Four hundred nanograms of trypsin was used to digest major slices and 200 ng to digest the minor ones.

#### 2.5.3. LC–MS/MS Analysis and Building a Spectral Library

To build a spectral library for the SWATH-MS analysis, by means of an Ekspert nanoLC 425 (Eksigent, Dublin, CA, USA), 2 µL of every digested sample was pooled and 5 µL of the pool was processed. Five microliters of peptide mixtures were loaded onto a trap column (3µ C18-CL, 350 µm × 0.5 mm; Eksigent) and desalted with 0.1% TFA at 5 µL/min for 3 min. The peptides were then loaded onto an analytical column (3µ C18-CL 120 Ᾰ, 0.075 × 150 mm; Eksigent) balanced in 5% acetonitrile (ACN) 0.1% formic acid (FA). Elution was carried out with a linear gradient of 7–40% B in A for 60 min (A: 0.1% FA; B: ACN, 0.1% FA) at a flow rate of 300 µL/min. The peptides were analyzed in a nanoESI qQTOF mass spectrometer (6600 plus TripleTOF, ABSCIEX, Framingham, MA, USA).

The samples were ionized in a Source Type: Optiflow < 1 µL Nano applying 3.0 kV to the spray emitter at 175 °C. The analyses were carried out in data-dependent mode. Survey MS1 scans were acquired from 350 to 1400 *m*/*z* for 250 ms. Quadrupole resolution was set to ‘LOW’ for the MS2 experiments, which were acquired at 100–1500 *m*/*z* for 25 ms in ‘high sensitivity’ mode. The following switch criteria were used: charge: from 2+ to 4+; minimum intensity; 250 counts per second (cps). Up to 100 ions were selected for fragmentation after each survey scan. Dynamic exclusion was set to 15 s. All the spectra obtained were combined and used to generate the reference spectral ion library as part of the SWATH-MS analysis.

#### 2.5.4. SWATH–MS Analysis

For every mixture of digested peptides, 3 µL of peptide mixture sample was loaded onto a trap column as described in the previous paragraph. After ionization, the TripleTOF was operated in SWATH mode, in which a 0.050 s TOF MS scan from 350 to 1250 *m*/*z* was first performed. Subsequently, 0.080 s product ion scans in 100 variable windows from 400 to 1250 *m*/*z* were acquired throughout the experiment. The total cycle time was 2.79 s. The individual SWATH injections were randomized to avoid bias in the analysis.

### 2.6. Identification and Quantification of Proteins

After DDA LC-MS/MS, the WIFF data files were processed using the ProteinPilot v5.0 search engine (SCIEX, Framingham, MA, USA). The Paragon algorithm (SProteinPilot) was employed to search against the SwissProt database (200,601, 562,246 proteins). Searches were made with trypsin specificity and IAM Cys-alkylation, without taxonomy restriction, and the search effort was set to Rapid with FDR analysis. Among the proteins identified by LC-MS/MS in DDA mode in each sample, only those showing ProteinPilot unused scores above 1.3 (>95% confidence threshold) and a false discovery rate (FDR) lower than 1% were considered significant and included in the analysis. Regarding the SWATH analysis, the proteins identified by means of the DDA analysis were quantified using the PeakView 2.2 (SCIEX) software for normalized label-free quantification intensity data. The spectral library generated was used as a database in the Peak View 2.2 software for the SWATH analysis, and peaks from SWATH runs were extracted with a peptide confidence threshold of 95% and FDR lower than 1%. It was not set to a minimum number of peptides for quantitation. The quantitated protein areas were normalized with Marker View 1.3 (SCIEX) by the total area sum for differential expression analysis.

### 2.7. Bioinformatic Analysis

Proteins with |log2FC| > 1 and *p*-value < 0.05 according to the Welch t-test were selected as differentially expressed across the three groups. The biological annotation and graphical representation were performed with R (v4.2.0) libraries STRINGdb [22], ggplot2 [23] and ComplexHeatmat [24]. For the heatmap, the matrix with the areas was log2-normalized and scaled with the R function scale. Hierarchical clustering was performed on both rows and columns in order to compute the dendrograms, with complete linkage and Euclidean distance metrics [25] to obtain a detailed explanation on these parameters. Pathway enrichment for COVID-19-exclusive DEPs was performed with Reactome annotation [26]. Annotated Gene Ontology (GO) terms with an FDR < 0.01 were filtered with Revigo [27] to remove redundant terms.

## 3. Results

### 3.1. SARS-CoV-2 Detection in Colostrum

Colostrum samples from mothers with COVID-19 tested negative for SARS-CoV-2 in RT-qPCR analysis, which excluded the risk of direct viral transmission through milk.

### 3.2. Proteomic Profile of the Colostrum Samples

Proteomic expression profile of colostrum samples was analyzed in women with COVID-19 (*n* = 3), women recently recovered from the infection (*n* = 4), and noninfected women (*n* = 5). A total of 301 abundant proteins were detected in all the colostrum samples analyzed. The total list of proteins identified for each sample analyzed is shown in Appendix A There were 53 proteins differentially expressed between at least two of the three groups (Figure 2).

We analyzed a heatmap of the differentially expressed proteins across all three groups. A similar expression pattern is observed in the samples belonging to the same group and different from those of the other two groups. The recovered group is observed as an intermediate group between the other two (Figure 3).

### 3.3. Discriminant Analysis

The discriminant analysis (DA) suggested a distinction between the control group and the other two groups. This result suggests an alteration in colostrum proteome not only in active infection but also in recovered patients (Figure 4).

### 3.4. Alteration of the Proteome Profile in Colostrum from COVID-19 Patients

Our results showed that there were significant differences in the expression of several proteins during SARS-CoV-2 infection when compared to noninfected controls and recovered patients. Thus, 19 proteins were upregulated and 7 proteins were downregulated in the comparison of COVID-19 versus control samples, whereas 18 were upregulated and 7 were downregulated when we compared the COVID-19 group to the recovered group (Figure 2). Moreover, 11 proteins were potential biomarkers of active SARS-CoV-2 infection in colostrum: ACTN1, CD36, FAM3B, GPRC5B, IGHA2, IGK, PLTP, RAC1, SDCBP and SERPINF1 showed higher levels, while PSAP was downregulated (Figure 2). The analyses of the pathways of these potential biomarkers revealed their implication in platelet functions and syndecan-4-mediated signal events (Figure 5).

A simplified Gene Ontology (GO) annotation provided a detailed view of the functional classes of the differentially expressed proteins. There were 30 enriched GO terms in the comparison between the COVID-19 colostrum and control colostrum groups (Figure 6). Enrichment of proteins present in the extracellular space and vesicles was observed in the COVID-19 samples, indicating that most of the proteins are secreted. Importantly, the proteins related to leukocyte-mediated immunity, humoral immunity and platelet activity were increased in the mothers affected by COVID-19.

A prediction of the interactions of each differentially expressed colostrum protein was made to see what their interactions were like (Figure 7). The interesting node proteins due to the number of interactions were A2M, GAPDH, GC and HP. A2M (alpha-2-macroglobulin) showed the highest number of interactions (nine), including CFI, ENSP00000450540, GAPDH, GC, HP, SDCBP, SERPINF1, SERPINA3 (GIG25) and SHBG. GAPDH (glyceraldehyde phosphate dehydrogenase) had seven interactions with other proteins. It is related to ACTN1, A2M, CRYZ, FTL, LPL, RAC1 and TALDO1. GC (vitamin-D binding protein) had six interactions: A2M, CFI, ENSP00000450540, HP, SERPINA3 and SHBG; and HP was related to six proteins: A2M, ENSP00000450540, GC, LPL, PLTP and SERPINA3.

On the other hand, we also found some upregulated proteins in the control colostrum when compared to the colostrum of women with COVID-19 (Figure 8). The kappa casein (CSN3) content was reduced nearly tenfold in COVID-19 samples. Other downregulated proteins were CD14, a membrane receptor of mononuclear phagocytes (MNPs), and sex hormone-binding globulin (SHBG).

### 3.5. The COVID-19-Recovered Group Differs from the Control and COVID-19 Groups

The COVID-19-recovered group is different from both the control and COVID-19 groups. In women that have recently overcome the infection, the colostrum still did not behave as noninfected and there were significant differences in the expression of several proteins when compared to the control group. Thus, 14 proteins were upregulated and 9 were downregulated in the comparison with the control samples. On the other hand, 7 proteins were upregulated and 18 downregulated in the comparison with the COVID-19 group. Five proteins were potential biomarkers of this group: ZG16B and DCS2 were upregulated, whereas RNASCT2, A1BG and TUBA1C were downregulated (Figure 2).

A simplified Gene Ontology (GO) annotation provided a detailed view of the functional classes of the differentially expressed proteins. Comparing the COVID-19-recovered and control groups, there were 12 enriched GO terms. Enrichment of proteins related to exosomes, vesicles, blood microparticles and platelets was observed (Figure 9).

A prediction of the interactions of each differentially expressed colostrum protein was made to see what their interactions were like (Figure 10). The interesting node proteins by number of interactions were A2M, GC, HP, ITIH4, KNG1 and A1BG. ITIH4 (interalphatrypsin inhibitor heavy chain H4) had six interactions with other proteins. It was related to HP, GC, A2M, THBS1, KNG1 and A1BG. KNG1 (kininogen-1) showed six interactions, including CSN3, ITIH4, HP, GC, A2M and A1BG.

A volcano plot was also performed to compare the rising and falling proteins in the COVID-19-recovered group versus the control group (Figure 11). The proteins that appear in red decreased concerning the control group, as was the case with sex hormone-binding globulin (SHBG), HSPA8 and RNASET2. On the other hand, the proteins shown in blue were increased in the recovered group, like THBS1, an antiangiogenic protein, or KNG1, a proinflammatory protein [28].

## 4. Discussion

Human milk is known as an important source of bioactive compounds that promote development and health in newborns. Moreover, it plays a main role on protection from infections and development of immunity [29]. These substances are mainly proteins and are concentrated in the colostrum, the milk produced at the first lactation moments. During an infection either in the mother or the infant, the change in human milk composition reflects both maternal and infant health conditions [30,31] and provides the suckling infant with protection against diseases related to intestinal and respiratory infections [32]. However, since the outbreak of the COVID-19 pandemic, women with this disease have been significantly concerned about breastfeeding. For this reason, the first objective of this study was to analyze the presence of SARS-CoV-2 in the colostrum. Based on the RT-qPCR results, the colostrum samples from mothers with active COVID-19 infection tested negative for SARS-CoV-2; this excluded the risk of viral transmission through milk. To the best of our knowledge, only six studies have analyzed the presence of SARS-CoV-2 in colostrum [14,33,34,35,36,37]. Of these, only two have detected presence of viral RNA in any of the samples, although the sample size was small in these studies [34,37]. This viral presence might be due to contamination during sampling, either through the skin of the breasts or due to nonuse of masks during sampling [38].

The second objective was to characterize the protein profile in colostrum collected from mothers with COVID-19, post-infection recovered and noninfected donors using a new methodology: sequential window acquisition of all theoretical fragment ion spectra-MS (SWATH-MS). A total of 301 abundant proteins were identified in all the colostrum samples. Many of the proteins detected in the samples have been previously described in colostrum, for instance, complement factor C3 and C4; haptoglobin; hemopexin; kallikrein 6; kappa casein; lipoprotein lipase; mucin 1, 4 and 5B; osteopontin; and lactoferrin [39,40].

The results obtained from the DA suggested an alteration in milk proteome in patients with active infections when compared to the control group. Our results are in line with a previous study where dimensional reduction analysis (principal component analysis (PCA)) was used to analyze the colostrum proteome of COVID-19 patients and showed a clear separation from the control group [14]. Moreover, in another study, serum proteins from COVID-19 patients analyzed by means of uniform manifold approximation and projection (UMAP) showed certain degree of separation between the groups (healthy individuals, mild, severe and sick COVID-19 patients but SARS-CoV-2-negative) [41]. However, there are no previous studies analyzing human colostrum in recovered mothers. In our work, we suggest an alteration in colostrum proteome not only in active infections but also in recovered patients.

The comparison between the COVID-19 colostrum and control colostrum groups showed 30 enriched GO terms. Enrichment of proteins present in the extracellular space and vesicles was observed in the COVID-19 samples, indicating that most of the proteins are secreted. Exosomes (membrane macrovesicles secreted by cells and found in many biological fluids) facilitate immune development of the infant [40]. Importantly, proteins related to leukocyte-mediated immunity and humoral immunity are increased in mothers affected by COVID-19. Moreover, proteins related to platelet activity are upregulated; this fact may be related to thrombosis and strokes in COVID-19 patients [42,43,44]. Analysis of the interactions of the differentially expressed colostrum proteins showed the following proteins: A2M, GAPDH, GC and HP. A2M has activities both as a protease inhibitor and as a protein carrier, hence the multiple interactions [45]. GAPDH (glyceraldehyde phosphate dehydrogenase) had seven interactions with other proteins. It was related to ACTN1, A2M, CRYZ, FTL, LPL, RAC1 and TALDO1. This might be because it is a protein involved in cellular metabolism, participating in glycolysis and vesicle trafficking during exocytosis. GC (vitamin D-binding protein) had six interactions: A2M, CFI, ENSP00000450540, HP, SERPINA3 and SHBG. It is a protein involved in vitamin D transport and storage and is also involved in macrophage activation. This protein has previously been proposed as a prognostic biomarker in patients with SARS-CoV-2 infection [46]. HP (haptoglobin) was related to six of the proteins: A2M, ENSP00000450540, GC, LPL, PLTP and SERPINA3. It is interesting to note that patients with less severe SARS-CoV-2 infections had higher HP levels in their blood serum than those intubated or deceased [47]. In our study, HP was upregulated in the colostrum of the COVID-19 group, coinciding with the mild clinical symptoms of these mothers (Table 2).

As already mentioned, an increase in immune-related proteins and protein trafficking is observed. IgA is the main immunoglobulin in human milk, specifically in colostrum, conferring immunity to the infant. Components of this multimer are clearly overexpressed in the colostrum of COVID-19 patients, like the alpha heavy chain (IGHA2) or immunoglobulin kappa proteins. RAC1 is involved in neutrophils’ inflammatory recruitment and chemotactic response [48]. SDCBP is involved in immunomodulation and its overexpression inhibits HIV-mediated cell fusion and HIV-1 production. On the other hand, SDCBP depletion increases HIV-1 entrance during infection [49]. Furthermore, SDCBP has been identified as the target for the prediction and subsequent treatment of COVID-19 because this protein is involved in the positive regulation of class II HLA, which is significantly increased in patients with less severe forms of the disease [50]. FAM3B is a pancreas-derived cytokine involved in glucose homeostasis and whose expression is induced by insulin and proinflammatory cytokines. Higher FAM3B serum levels in COVID-19 patients have been linked to worse infection outcomes [51]. PLTP and LPL are involved in the lipolysis of triglycerides and are known to decrease in mature milk, also reducing the influx of free fatty acids (FFAs) from blood lipid droplets to the milk [52]. In our case, an increase in PLTP and a decrease in LPL in COVID-19 samples were observed; however, the possible impacts on lipid composition are unknown. Other authors did not find significant changes in lipid production when colostrum from COVID-19 patients and healthy women were compared [14]. ACTN1 is a cytoskeletal protein previously described in milk and involved in the secretion of the milk fat globule membrane (MFGM) [53]. SERPINs are protease inhibitors with direct involvement in fibrinolysis and SARS-CoV-2 infection. COVID-19 severity may be associated with an increase in the levels of SERPIN proteins. In our work, SERPINF1 is upregulated and has been previously suggested as a biomarker of prognosis for infection severity [54].

On the other hand, the kappa casein (CSN3) content was reduced nearly tenfold in COVID-19 samples when compared to control samples. CSN3 is critical for lactation, as shown in Csn3^−/−^ mice [55]. In the absence of CSN3, the remaining caseins are retained in the alveolar lumina of the mammary gland and cannot be ejected. Thus, the lower CSN3 levels in the colostrum from patients infected with SARS-CoV-2 might also explain the aberrant expression of other milk proteins. Another downregulated protein (CD14) is a membrane receptor of mononuclear phagocytes (MNPs), inducing phagocyte hyperactivation in the presence of inflammatory lipids, like bacterial lipopolysaccharide (LPS), thus participating in innate immunity [56]. Previous research has shown downregulation in proteins related to neutrophil degranulation in milk in response to COVID-19 [14], but CD14 downregulation would also mean that there are fewer MNPs in the colostrum of COVID-19 patients, perhaps as a result of recruiting to COVID-19-infected tissues. On the other hand, CD14 inhibition has shown to be beneficial in attenuating the detrimental effect of complement activation and modulation of the cytokine storm in fulminant sepsis patients [57]. Sex hormone-binding globulin (SHBG) was also higher in the control group, which is in contrast with previous reports of increased SHBG serum levels in male COVID-19 patients [28]. In future studies, it would be interesting to compare serum proteins to colostrum proteins in the same patients.

To the present day, only two studies have analyzed the proteome of colostrum from patients with COVID-19 [14,35]. They examined colostrum samples collected on the third or fourth postpartum days. Some authors consider transition milk as from 72 h postpartum [9]. Therefore, our study was able to reach more accuracy in analyzing colostrum because the samples were taken 20 ± 5.9 h postpartum. In Zhao et al. (2020), 88 differentially expressed proteins (DEPs) related to inflammatory processes, immune response and metabolism were found [14]. However, contrary to our results, they indicate that numerous human milk proteins involved in the immune response were downregulated in response to COVID-19, like markers for neutrophil degranulation (CD44 and complement factor properdin (CFP)) and leukocyte migration. In the same way, they found downregulated markers for platelet degranulation, as opposed to our results [14].

Another recent report, with data from six patients and ten healthy controls, found a total of 340 DEPs. Upregulated proteins were involved in vesicle-mediated transport, leukocyte-mediated immunity and regulation of the humoral immune response [35]. These results are totally in line with our findings, where we detected a general increase in proteins involved in immune processes, like Igs and proteins involved in leukocyte migration or neutrophil degranulation. On the other hand, other authors have examined serum from patients with COVID-19, observing that the affected proteins belonged mainly to three metabolic pathways: macrophage function, activation of the complement system and platelet degranulation [41].

Moreover, according to our data, some nutritional components of human colostrum are reduced in the colostrum of COVID-19 patients, including kappa casein (CSN3). This result is in line with the research study by Guo et al. (2022), where caseins were found to be downregulated in these patients [35].

Thus, the colostrum from SARS-CoV-2 infected mothers may provide defense against the coronavirus to neonates, similarly to the milk of HIV-affected mothers [11]. However, these results are in opposition to what has been previously described by Zhao et al. (2020) [14], who concluded that human milk from mothers with COVID-19 was deficient in immune-related components. Be it one way or another, it is still unclear if the immune defense provided by this human milk to neonates is substantially different from those breastfed by noninfected mothers, excluding the protective effect against coronavirus. This is also seen in other diseases, including HIV. Inhibition of HIV transmission through human milk is partly due to tenascin-C, a protein present in human milk. This protein interacts with the envelope domain of the virus and neutralizes it [17].

On the other hand, this study is the first one that analyzes colostrum from recovered patients. In women that have recently overcome the infection, their colostrum still had alterations and there were significant differences in the expression of several proteins when compared to the control group. Thus, enrichment of proteins related to exosomes, vesicles, blood particles and platelets was observed. Some of these groups of proteins are altered in COVID-19 colostrum, like vesicles and platelet proteins. However, we did not find any change in the group of proteins related to immunity. On the other hand, the analysis of the interactions of each differentially expressed colostrum protein showed the following proteins: A2M, GC, HP, ITIH4, KNG1 and A1BG. ITIH4 (interalphatrypsin inhibitor heavy chain H4) had six interactions with other proteins. This protein acts as a protease inhibitor and is related to inflammation. Regarding this, it has been observed that its concentration is increased in the serum of COVID-19 patients [58]. KNG1 (kininogen-1) showed six interactions. This protein participates in blood coagulation and has been found in higher amounts in the serum of COVID-19 patients [59].

The volcano plot showed some proteins that decreased concerning the control group, as was the case with sex hormone-binding globulin (SHBG), HSPA8, which confers protection of the proteome against stress [60], and RNASET2, which degrades the RNAs of microorganisms and plays a role in the innate immune response [61]. On the other hand, the proteins shown in blue were increased in the recovered group, for instance, THBS1, an antiangiogenic protein, which is increased in the serum of asymptomatic COVID-19 patients, as well as other proteins related to endothelial dysfunction [62] or KNG1, a proinflammatory protein [63].

In summary, changes in colostrum proteins were observed between the three groups studied. The samples from recovered patients presented an intermediate behavior between the control group and the COVID-19 group. Moreover, a gradual difference is observed among the samples from recovered patients. The RECOV9 sample was taken 6 months after the infection, unlike the other samples, which were taken 3 months after the infection. We observed that RECOV9 is more similar to the control group samples than others (RECOV10, RECOV13 and RECOV14). This observation suggests gradual normalization on colostrum composition in recovered patients. Thus, in a longitudinal study, it was described that the changes found in the proteomic profile of colostrum in COVID-19 patients disappear in mature human milk [35].

The results of this study suggest that the colostrum from mothers with COVID-19 confers immune benefits to infants. Future studies will allow for understanding the function of the proteins detected in newborns. Some of these proteins might be used as potential biomarkers of COVID-19 infection or severity. Furthermore, taking human milk as a biological reference model, experimental studies might reveal the application of some of these molecules in treatment of the disease.

## 5. Conclusions

Our findings demonstrate that colostrum is not a vehicle for mother-to-child SARS-CoV-2 transmission. On the other hand, the proteomics analyses by means of SWATH-MS revealed the significant alterations of numerous proteins associated with COVID-19. Notably, differentially expressed proteins in the colostrum of COVID-19 mothers are mainly related to immunity, inflammatory response and protein transport. These changes were probably a reflection of the mother’s whole-body physiological responses to SARS-CoV-2 infection and may contribute to establishing immune defense in the early life of their neonates. Moreover, for the first time, the proteomic profile of colostrum in recovered patients was described, showing a different behavior when compared to controls and mothers with active infections. Furthermore, this is the first study [35] related with COVID-19 in which the colostrum was taken in the first 24 h after delivery. There are other similar studies, but they used transition milk because samples were collected in the third or fourth day postpartum [14,35].

On the other hand, the reduced number of colostrum samples, due to restricted access to hospitals during the first period of the COVID-19 pandemic, might be considered as a limitation of our work. Further studies would be necessary to carry out an in-depth analysis of the detected biomarkers and their function in colostrum.

## Figures and Tables

**Figure 1 children-10-01423-f001:**
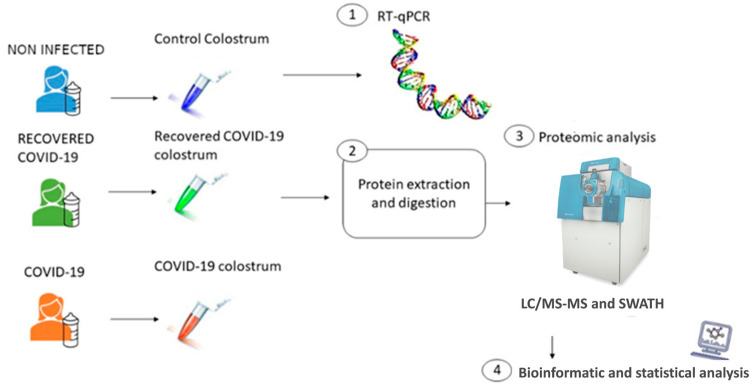
Schematic representation of the study design. Colostrum samples were collected from active COVID-19 women (*n* = 3), recovered (*n* = 4) and noninfected women (*n* = 5). The workflow for processing the omics study was as follows: Firstly, RT-qPCR was performed to determine the presence or absence of viral RNA. Secondly, protein extraction and digestion for SWATH-MS and proteomic analysis was performed. Finally, proteomic results were analyzed using statistical and bioinformatics analysis.

**Figure 2 children-10-01423-f002:**

Venn diagram showing the differentially expressed proteins between the control, recovered and COVID-19 groups. Overexpressed proteins unique to each group (potential biomarkers) are shown in color.

**Figure 3 children-10-01423-f003:**
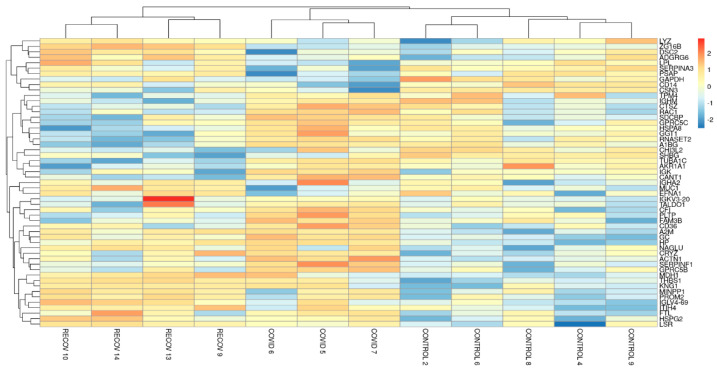
Heatmap of the differentially expressed proteins among the three groups. Red represents higher expression and blue represents lower expression. The Y-axis shows the name of the different proteins analyzed and the X-axis shows the group to which the sample belongs.

**Figure 4 children-10-01423-f004:**
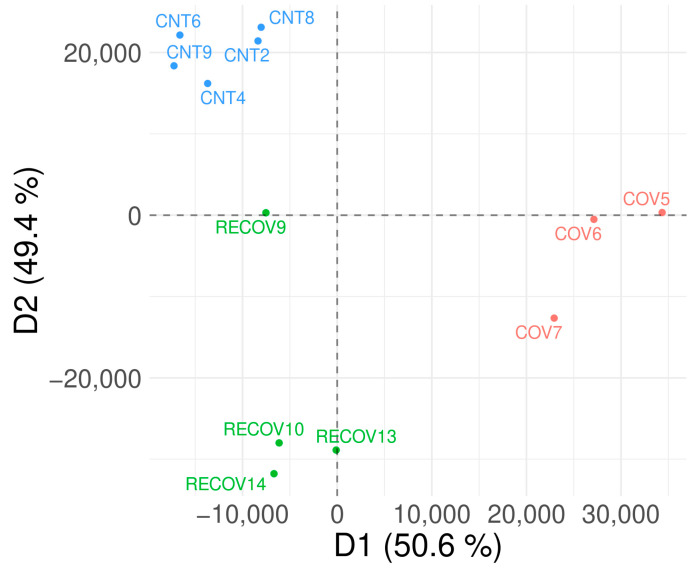
DA analysis shows that the three groups studied are clearly differentiated. Control samples are shown in blue in the top left, COVID-19 samples are shown in red in the bottom right and recovered COVID-19 samples are shown in green in the bottom left. The axes represent the two most important dimensions in the discrimination of sample groups. Dimension 1 (D1) explains 50.6% of the differences while dimension 2 explains the remaining 49.4%.

**Figure 5 children-10-01423-f005:**
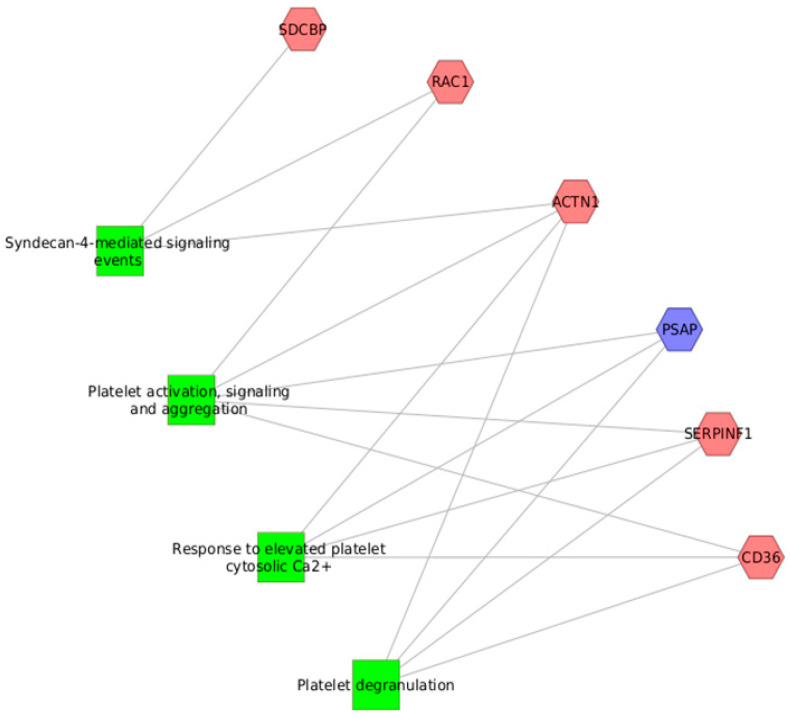
Pathways analyses of potential biomarkers of an active SARS-CoV-2 infection in colostrum. In light red: overexpressed proteins. In light blue: underexpressed protein. The pathways are shown as green rectangles.

**Figure 6 children-10-01423-f006:**
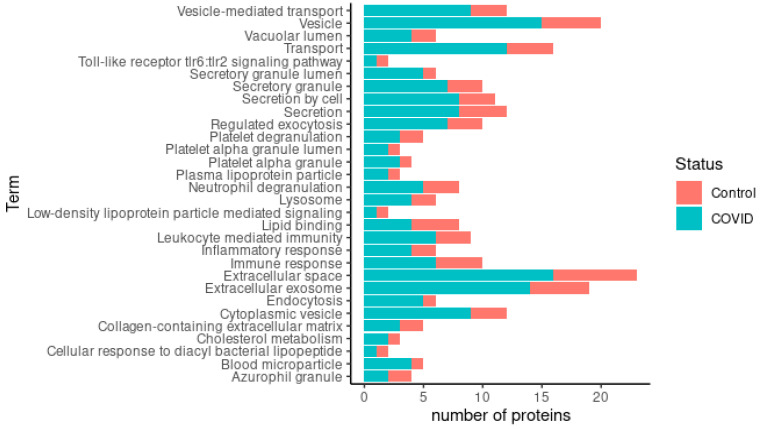
Representation of enriched GO terms in COVID-19 group in comparison to control group. The total number of proteins associated with each term equals the total length of the bar. The most abundant proteins in colostrum from control group are represented by the red part and the most abundant proteins in colostrum from COVID-19 females are represented by the turquoise blue part.

**Figure 7 children-10-01423-f007:**
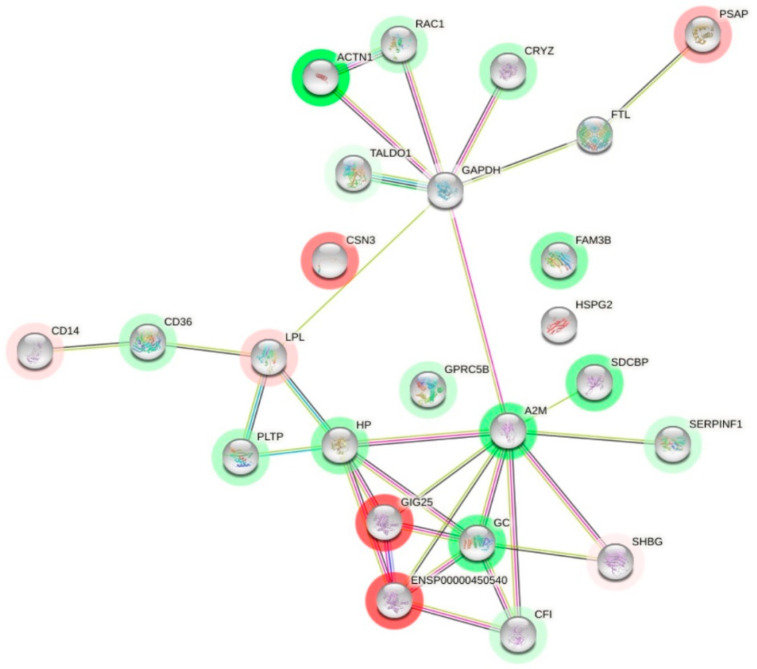
Interaction network of differentially expressed proteins in COVID-19 group. Proteins whose expression decreases with COVID-19 infection are shown in red and those whose expression increases are shown in green. Black lines indicate co-expression. Proteins that have appeared in the same articles by making a bibliographic search are shown in green line. Pink lines indicate proteins whose interaction has been experimentally determined in humans. Blue line indicates that it has been documented in a database.

**Figure 8 children-10-01423-f008:**
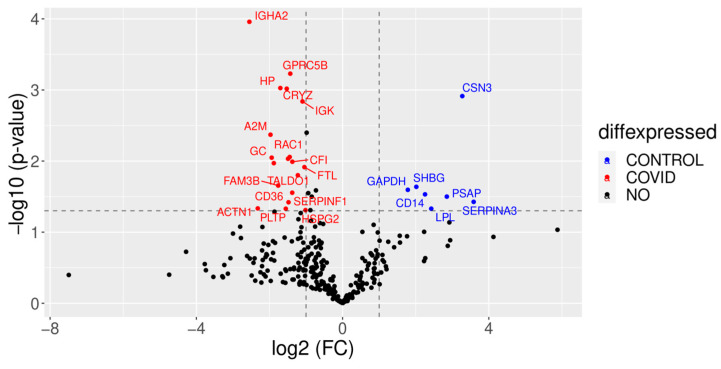
A volcano plot showing proteome changes between COVID-19 colostrum and control colostrum groups. Red shows the proteins that are expressed in lower quantities in the control than in the COVID-19 group, while blue shows the proteins that are expressed in higher quantities in the control group than in the COVID-19 group. The Y-axis represents the robustness of the results; a higher value indicates a higher probability.

**Figure 9 children-10-01423-f009:**
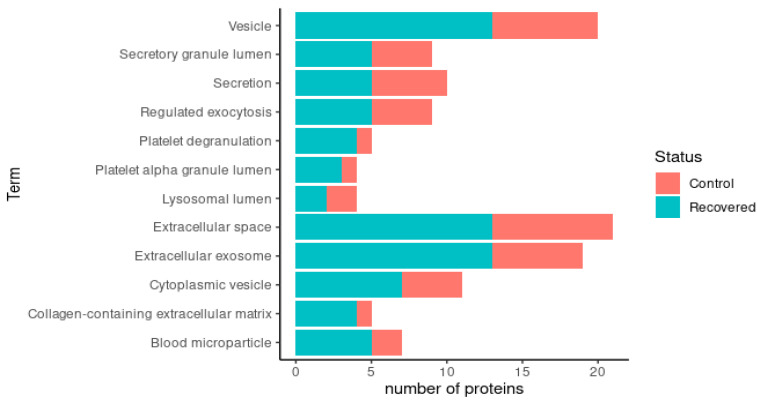
Representation of enriched GO terms in the recovered group in comparison to control. The total number of proteins associated with each term equals the total length of the bar. The most abundant proteins in colostrum from control women are represented by the red part and the most abundant proteins in colostrum from recovered women are represented by the turquoise blue part.

**Figure 10 children-10-01423-f010:**
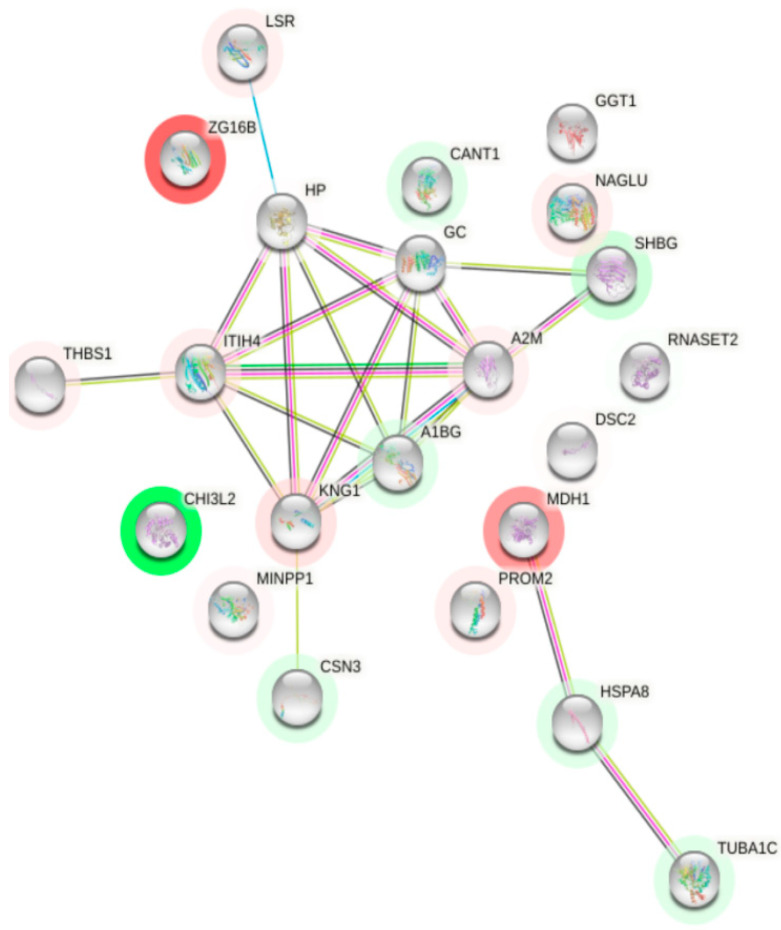
Interaction network of differentially expressed proteins in recovered group. Proteins whose expression decreases in recovered group are shown in red and those whose expression increases are shown in green. Black lines indicate co-expression. Proteins that have appeared in the same articles by making a bibliographic search are shown in line green. Pink lines indicate proteins whose interaction has been experimentally determined in humans. Blue indicates that it has been documented in a database.

**Figure 11 children-10-01423-f011:**
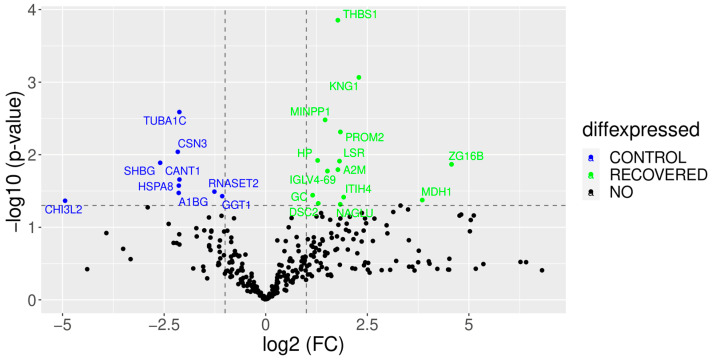
A volcano plot showing proteome changes between recovered colostrum and control colostrum groups. Blue shows the proteins that are expressed in lower quantities in the recovered than in the control group, while green shows the proteins that are expressed in higher quantities in the recovered group than in the control group. The Y-axis represents the robustness of the result, a higher value indicates a higher probability.

**Table 1 children-10-01423-t001:** Participants inclusion and exclusion criteria.

Criteria	Characteristics
Inclusion criteria	Full-term pregnancy
	Absence of infection during gestation period (except SARS-CoV-2 infection in recovered group)
	Absence of infection at the moment of delivery (except SARS-CoV-2 infection in COVID-19 group)
Exclusion criteria	Preterm pregnancy
	Infectious disease during gestation (except SARS-CoV-2 infection in recovered group)
	Infectious disease at the moment of the delivery (except SARS-CoV-2 infection in COVID-19 group)
	Immunocompromised mothers
	Mothers whose babies are in Neonatal Intensive Care Unit
	Caesarea

**Table 2 children-10-01423-t002:** Characteristics of colostrum donors.

Characteristics	Noninfected (*n* = 5)	Recovered COVID-19 (*n* = 4)	COVID-19 (*n* = 3)
Maternal age: mean years (SD)	27.37 (7.86)	33.53 (1.89)	31.85 (9.06)
Ethnicity	Caucasian	Caucasian	Caucasian
Infectious diseases (apart from COVID-19)	No	No	No
Other complications occurred during pregnancy (diabetes, preeclampsia, anemia, etc.)	ICP * (donor CNT6)	No	No
Women vaccinated against SARS-CoV-2	-	2 Donors (RECOV10 and 13)	-
Number of vaccine doses	-	2	-
Vaccine type (adenovirus-based/mRNA)	-	mRNA	-
COVID-19 detection: mean days before delivery (SD)	-	123.25 (34.29)	1
PCR by nasal swabs at the delivery day (negative/positive)	Negative	Negative	Positive
Gravidity: mean (SD)	2 (1.2)	2.75 (1.5)	2.66 (2.08)
Type of delivery (vaginal/caesarean)	Vaginal	Vaginal	Vaginal
Birth week: mean (SD)	38.88 (1.54)	39.61 (0.7)	37.85 (0.66)
Collection of colostrum postdelivery: mean hours (SD)	19.2 (6.6)	24	16 (6.9)
Severity of the COVID-19 infection (mild/severe)	-	Mild	Mild
Mild symptoms (% of mother with at least one mild symptom: myalgia, headache, anosmia, low-grade fever)	-	75%	66%
Severe symptoms (% of mother with severe symptoms: trouble breathing, persistent pressure or pain in the chest, confusion, pale, grey or blue-colored skin	-	0%	0%

* ICP: intrahepatic cholestasis of pregnancy.

**Table 3 children-10-01423-t003:** Real-time RT-qPCR panel primers and probes.

Name	Description	Oligonucleotide Sequence (5′ > 3′)
2019-nCoV_N1-F	2019-nCoV_N1 forward primer	GAC CCC AAA ATC AGC GAA AT
2019-nCoV_N1-R	2019-nCoV_N1 reverse primer	TCT GGT TAC TGC CAG TTG AAT CTG
2019-nCoV_N1-P	2019-nCoV_N1 probe	FAM-ACC CCG CAT TAG GTT TGG TGG ACC-BHQ1
RP-F	RNase P forward primer	AGA TTT GGA CCT GCG AGC G
RP-R	RNase P reverse primer	GAG CGG CTG TCT CCA CAA GT
RP-P	RNase P probe	FAM-TTC TGA CCT GAA GGC TCT GCG CG-BHQ-1
E_Sarbeco_F1	E_Sarbeco forward primer	ACAGGTACGTTAATAGTTAATAGCGT
E_Sarbeco_R2	E_Sarbeco reverse primer	ATATTGCAGCAGTACGCACACA
E_Sarbeco_P1	E_Sarbeco probe	FAM-ACACTAGCCATCCTTACTGCGCTTCGBBQ

## Data Availability

Not applicable.

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
