# Peer review of "Colostrum Features of Active and Recovered COVID-19 Patients Revealed Using Next-Generation Proteomics Technique, SWATH-MS"

_children, 2023, doi:10.3390/children10081423_

Round 1

Reviewer 1 Report

Dear Authors,

I congratulate you for your work. I read the manuscript and this are my recommendations to improve the article.

The abstract should include details about the methodology, the number or analysed samples and more accurate conclusions.

The introduction is very well detailed. I should say that is to extensive but it support the purpose of the manuscript.

The first concern in the result section is given by the images. Are they original? Do you have any copyright issue? Please can you solve that problem?

Another issue and the main problem of the manuscript is the reduced number of the subjects. Can you collect data from more women?

The results and discussions are well sustained but from my point of view the reduce number of subjects is a concern.

A separate conclusion section should be added.

Author Response

We would like to thank the reviewers for their detailed comments and suggestions for our manuscript entitled " Colostrum Features of Active and Recovered COVID-19 Patients Revealed by Next Generation Proteomics Technique, SWATH-MS".

We believe that the modifications introduced have improved the manuscript. Below, you will find a point-by-point description of how each comment was addressed in the manuscript. Original reviewer comments in boldface and responses in regular typeface.

We hope this new version will be suitable for publication.

Reviewer 1

Dear Authors,

I congratulate you for your work. I read the manuscript and this are my recommendations to improve the article.

We would like to thank the reviewer for his/her valuable comments. We really think that they have improved the manuscript.

The abstract should include details about the methodology, the number or analysed samples and more accurate conclusions.

The abstract has been modified following the reviewer suggestions (lines 32 to 35 and 41 to 44).

The introduction is very well detailed. I should say that is to extensive but it support the purpose of the manuscript.

We are thankful to the reviewer comments’ and we hope that the new version will be suitable for publication.

The first concern in the result section is given by the images. Are they original? Do you have any copyright issue? Please can you solve that problem?

All the images presented in this manuscript are original. Therefore, there is no copyright infringement.

Another issue and the main problem of the manuscript is the reduced number of the subjects. Can you collect data from more women? The results and discussions are well sustained but from my point of view the reduce number of subjects is a concern.

As the reviewer points the number of samples might be small. This is due to the limited access to this kind of samples in the hospital during COVID-19 restrictions (in the early moments of the pandemic) due to biosecurity protocols. Moreover, today it is very difficult to collect data from unvaccinated women and with active infection. Furthermore, in our hospital it is almost impossible to find mothers for the control group with no previous vaccination.

Data from other proteomic experiments on human milk of COVID-19 patients are not abundant and sample size is modest. For example, Zhao and coworkers (2020) showed some conclusions from a smaller sample size (n=4 COVID patients and n=2 healthy women as a control), identifying 88 DEPs with a |FC| > 1 and a p-value (unadjusted) < 0.05. Other proteomic studies of breast milk have been published with few samples (Liao et al., 2011 (n=5); Chruscicki et al., 2017 (n=1); Liao et al., 2017 (n=4); Picariello et al., 2019 (n=1); Zhu et al., 2019 (n=6); Hahn et al., 2020 (n=3); Zhao et al., 2020 (n=1); Patel et al., 2021 (n=3); Dingess et al., 2021 (n=2); Milkovska-Stamenova et al., 2021 (n=1)).

Despite the limited number of samples, our work has many strengths. It is the first in describe the proteomic profile of colostrum in recovered COVID-19 patients. Moreover, it is the first study related with COVID-19 in which the colostrum was taken in the first 24h after delivery (other studies speak about colostrum but really they used transition milk). Thus, in the other two studies related with COVID-19 and colostrum, the colostrum milk from COVID-19 patients was collected the third or fourth day postpartum (Guo et al., 2022; Zhao Y et al., 2020). This human milk could be considered as transition milk and not colostrum (Zhu & Dingess, 2019). Furthermore, the use of SWATH-MS technique has shown deep proteome coverage coupled to higher performance and reproducibility in previous studies, when compared to conventional data-dependent analysis mass spectrometry (DDA-MS) [20,21]. For these strengthens, in our opinion, the results of this study are relevant for the health professionals and scientific community .

A separate conclusion section should be added.

As suggested a conclusion section has been included (lines 554-570).

Reviewer 2 Report

The manuscript titled to Colostrum Features of Active and Recovered COVID-19 Patients Revealed by Next Generation Proteomics Technique, SWATH-MS was critically evaluated. I am in the opinion that this study is very important and was well-written. The only thing is that limitations and strenghts of the study can be added into conclusion section. Best regards,

Author Response

We would like to thank the reviewers for their detailed comments and suggestions for our manuscript entitled " Colostrum Features of Active and Recovered COVID-19 Patients Revealed by Next Generation Proteomics Technique, SWATH-MS".

We believe that the modifications introduced have improved the manuscript. Below, you will find a point-by-point description of how each comment was addressed in the manuscript. Original reviewer comments in boldface and responses in regular typeface.

We hope this new version will be suitable for publication.

Reviewer 2

The manuscript titled to Colostrum Features of Active and Recovered COVID-19 Patients Revealed by Next Generation Proteomics Technique, SWATH-MS was critically evaluated. I am in the opinion that this study is very important and was well-written. The only thing is that limitations and strenghts of the study can be added into conclusion section. Best regards.

We are very grateful for the suggestion made by the reviewer and as suggested we have included the strengths and limitations of the study into conclusion section (see lines 555-570).

Reviewer 3 Report

Number of cases included in the analysis should be noted in the results.

The number of cases should be clearly indicated in the results section and also in the table.

The number of cases in this study is too small to be evaluated (insufficient power).

It is not clear why only colostrum was included in the study (the effect of SARS-Cov-2 infection on biological material may not be limited to colostrum alone).

Why were other biological samples from lactating women not included in the analysis (milk components are also affected by serum proteins, etc.)?

Shouldn't the need for further study be added as a limitation?

Is the intra-day variation of the analyzed components and other influencing factors taken into account in the analysis? Is it better to regard the data as uncorrected data?

Author Response

We would like to thank the reviewers for their detailed comments and suggestions for our manuscript entitled " Colostrum Features of Active and Recovered COVID-19 Patients Revealed by Next Generation Proteomics Technique, SWATH-MS".

We believe that the modifications introduced have improved the manuscript. Below, you will find a point-by-point description of how each comment was addressed in the manuscript. Original reviewer comments in boldface and responses in regular typeface.

We hope this new version will be suitable for publication.

Reviewer 3

We would like to thank the reviewer for his/her valuable feedback. We have carefully revised the manuscript according to his/her comments and suggestions as indicated below.

Number of cases included in the analysis should be noted in the results. The number of cases should be clearly indicated in the results section and also in the table.

The number of cases has been included in the results section (lines 258-259) and also in table 2.

The number of cases in this study is too small to be evaluated (insufficient power).

As the reviewer points the number of samples might be small (it has been included as a limitation of our study in the limitations section). This is due to the limited access to this kind of samples in the hospital during COVID-19 restrictions (in the early moments of the pandemic) due to biosecurity protocols. Moreover, today it is very difficult to collect data from unvaccinated women and with active infection. Furthermore, in our hospital it is almost impossible to find mothers for the control group with no previous vaccination.

Data from other proteomic experiments on human milk of COVID-19 patients are not abundant and sample size is modest. For example, Zhao and coworkers (2020) showed some conclusions from a smaller sample size (n=4 COVID patients and n=2 healthy women as a control), identifying 88 DEPs with a |FC| > 1 and a p-value (unadjusted) < 0.05. Other proteomic studies of breast milk have been published with few samples (Liao et al., 2011 (n=5); Chruscicki et al., 2017 (n=1); Liao et al., 2017 (n=4); Picariello et al., 2019 (n=1); Zhu et al., 2019 (n=6); Hahn et al., 2020 (n=3); Zhao et al., 2020 (n=1); Patel et al., 2021 (n=3); Dingess et al., 2021 (n=2); Milkovska-Stamenova et al., 2021 (n=1)).

Despite the limited number of samples, our work has many strengths. It is the first in describe the proteomic profile of colostrum in recovered COVID-19 patients. Moreover, it is the first study related with COVID-19 in which the colostrum was taken in the first 24h after delivery (other studies speak about colostrum but really they used transition milk). Thus, in the other two studies related with COVID-19 and colostrum, the colostrum milk from COVID-19 patients was collected the third or fourth day postpartum (Guo et al., 2022; Zhao Y et al., 2020). This human milk could be considered as transition milk and not colostrum (Zhu & Dingess, 2019). Furthermore, the use of SWATH-MS technique has shown deep proteome coverage coupled to higher performance and reproducibility in previous studies, when compared to conventional data-dependent analysis mass spectrometry (DDA-MS) (Krasny et al., 2018; Ludwig et al., 2018). For these strengthens, in our opinion, the results of this study are relevant for the health professionals and scientific community.

It is not clear why only colostrum was included in the study (the effect of SARS-Cov-2 infection on biological material may not be limited to colostrum alone). Why were other biological samples from lactating women not included in the analysis (milk components are also affected by serum proteins, etc.)?

Effectively, as the referee say, the effects of SARS-Cov2 infection can be seen in other biological fluids. Previous proteomic studies have analyzed urine (Bi et al., 2022), saliva (Yanuchevish et al., 2022), or serum (Fraser et al. 2019; Shen et al., 2020; Geyer et al.,2021; Medjeral-Thomas et al., 2021; Yagzi et al., 2021; Beltrán-Camacho et al., 2022; Salonia et al., 2022). However, until now, only two studies in transition and mature milk have analyzed the proteome alterations during COVID-19 infection but not in colostrum (Zhao et al., 2022, Guo et al.,2022). That is why the study of colostrum was the unique objective of our study: for its importance in the transmission of the virus in the first days of the newborn and for its anti-inflammatory and anti-infective functions. Moreover, we were very interested in the detection of possible biomarkers of disease. In fact, some of the proteins detected in this study has been previously described in serum of COVID-19 patients: HP, FAM3B, ITIH4, KNG1, THBS1, etc. The implications of these results have been explained in the Discussion section.

References:

Bi X, Liu W, Ding X, Liang S, Zheng Y, Zhu X, Quan S, Yi X, Xiang N, Du J, Lyu H, Yu D, Zhang C, Xu L, Ge W, Zhan X, He J, Xiong Z, Zhang S, Li Y, Xu P, Zhu G, Wang D, Zhu H, Chen S, Li J, Zhao H, Zhu Y, Liu H, Xu J, Shen B, Guo T. Proteomic and metabolomic profiling of urine uncovers immune responses in patients with COVID-19. Cell Rep. 2022 Jan 18;38(3):110271. doi: 10.1016/j.celrep.2021.110271. Epub 2021 Dec 28. PMID: 35026155; PMCID: PMC8712267.

Beltrán-Camacho, L.; Eslava-Alcón, S.; Rojas-Torres, M.; Sánchez-Morillo, D.; Martinez-Nicolás, M. aP; Martín-Bermejo, V.; de la Torre, I.G.; Berrocoso, E.; Moreno, J.A.; Moreno-Luna, R.; et al. The Serum of COVID-19 Asymptomatic Patients up-Regulates Proteins Related to Endothelial Dysfunction and Viral Response in Circulating Angiogenic Cells Ex-Vivo. Molecular Medicine 2022, 28, doi:10.1186/S10020-022-00465-W.

Fraser, D.D.; Cepinskas, G.; Patterson, E.K.; Slessarev, M.; Martin, C.; Daley, M.; Patel, M.A.; Miller, M.R.; O’Gorman, D.B.; Gill, S.E.; et al. Novel Outcome Biomarkers Identified With Targeted Proteomic Analyses of Plasma From Critically Ill Coronavirus Disease 2019 Patients. Crit Care Explor 2020, 2, e0189, doi:10.1097/CCE.0000000000000189.

Geyer, P.E.; Arend, F.M.; Doll, S.; Louiset, M.; Virreira Winter, S.; Müller‐Reif, J.B.; Torun, F.M.; Weigand, M.; Eichhorn, P.; Bruegel, M.; et al. High-Resolution Serum Proteome Trajectories in COVID-19 Reveal Patient-Specific Seroconversion. EMBO Mol Med 2021, 13, doi:10.15252/EMMM.202114167.

Krasny, L.; Bland, P.; Kogata, N.; Wai, P.; Howard, B.A.; Natrajan, R.C.; Huang, P.H. SWATH Mass Spectrometry as a Tool for Quantitative Profiling of the Matrisome. J Proteomics 2018, 189, 11–22, doi:10.1016/J.JPROT.2018.02.026.

Ludwig, C.; Gillet, L.; Rosenberger, G.; Amon, S.; Collins, B.C.; Aebersold, R. Data‐independent Acquisition‐based <scp>SWATH</Scp> ‐ <scp>MS</Scp> for Quantitative Proteomics: A Tutorial. Mol Syst Biol 2018, 14, doi:10.15252/msb.20178126.

Medjeral-Thomas, N.R.; Troldborg, A.; Hansen, A.G.; Pihl, R.; Clarke, C.L.; Peters, J.E.; Thomas, D.C.; Willicombe, M.; Palarasah, Y.; Botto, M.; et al. Protease Inhibitor Plasma Concentrations Associate with COVID-19 Infection. Oxf Open Immunol 2021, 2, doi:10.1093/OXFIMM/IQAB014.

Salonia, A.; Pontillo, M.; Capogrosso, P.; Gregori, S.; Carenzi, C.; Ferrara, A.M.; Rowe, I.; Boeri, L.; Larcher, A.; Ramirez, G.A.; et al. Testosterone in Males with COVID-19: A 7-Month Cohort Study. Andrology 2022, 10, 34–41, doi:10.1111/ANDR.13097.

Shen, B.; Yi, X.; Sun, Y.; Bi, X.; Du, J.; Zhang, C.; Quan, S.; Zhang, F.; Sun, R.; Qian, L.; et al. Proteomic and Metabolomic Characterization of COVID-19 Patient Sera. Cell 2020, 182, 59, doi:10.1016/J.CELL.2020.05.032.

Yanuchevish OO, Mayev IV, Karton EA, Ostrovskaya IG. Proteomnyi analiz slyuny u patsientov s COVID-19 [Proteomic saliva assay in patients with COVID-19]. Stomatologiia (Mosk). 2022;101(4):34-37. Russian. doi: 10.17116/stomat202210104134. PMID: 35943498.

Yagci, S.; Serin, E.; Acicbe, Ö.; Zeren, M.İ.; Odabaşı, M.S. The Relationship between Serum Erythropoietin, Hepcidin, and Haptoglobin Levels with Disease Severity and Other Biochemical Values in Patients with COVID-19. Int J Lab Hematol 2021, 43 Suppl 1, 142–151, doi:10.1111/IJLH.13479.

Shouldn't the need for further study be added as a limitation?

As suggested by the reviewer this concern has been included in the conclusions as a limitation (lines 567-570).

Is the intra-day variation of the analyzed components and other influencing factors taken into account in the analysis? Is it better to regard the data as uncorrected data?

Although the data were uncorrected taking to account the collection day, several efforts have been made to address potential sources of bias in this trial. First of all, the selection bias was controlled by taking mothers using rigorous criteria to avoid confounding results. In Table 2, it is shown that the characteristics of donors were similar. In the second place, misclassification bias was avoided by performing a PCR in the moment of sample extraction and a rigorous study of the mother’s clinical history. Finally, bias due to systematic measurement error were controlled by several ways: routinely calibrating equipment (one per week), using controls in experiments (K562 digests), the person in charge of proteomic analysis worked with blinded samples regarding COVID-19 status, and finally, all samples were kept at -80ºC, being processed and analysed at the same time and under identical conditions. Moreover, to avoid bias associated with the acquisition in the equipment (SWATH analysis), the acquisition of the samples was randomized by blocks.

Round 2

Reviewer 1 Report

Dear Authors, 

I really appreciate that you followed the instructions from my report.

The manuscript is a valuable paper.

Reviewer 3 Report

No additional comments